# Barriers and facilitators to implementation of the Ethiopian national cancer control plan strategies: Implications for cervical cancer services in Ethiopia

Kemal Hussein[1]*, Francis Wafula[1], Getnet Mitike Kassie[2], Gilbert Kokwaro[1]

**1** Institute of Healthcare Management, Strathmore University, Nairobi, Kenya, **2** International Institute for Primary Healthcare–Ethiopia (IPHC-E), Addis Ababa, Ethiopia

* kemalahmedfenet@gmail.com

**Data Availability Statement:** All relevant data are within the paper and its Supporting Information files.

## Abstract

Following an upsurge in cervical cancer incidence and mortality, Ethiopia developed its first National Cancer Control Plan (NCCP) to support efforts toward the prevention and control of cancer. The NCCP outlines strategies for reducing the incidence of cancer through prevention, screening, early diagnosis, treatment, and palliative care. This study examined barriers and facilitators to the implementation of the NCCP using a qualitative approach. The study entailed doing key informant interviews and reviewing secondary data. Using customized topic guidelines, fifteen interviews were conducted covering a wide range of topics, including political commitment, priority setting, interagency cooperation, the role of evidence, citizen empowerment, and incentives. All interviews were recorded (with consent), transcribed in Amharic, and then translated into English for thematic analysis. Review of secondary data focused on establishing the NCCP's implementation status for HPV vaccination, cervical cancer screening, and treatment, and strategic links to five other national policy documents centered on public awareness, cervical cancer services, HPV immunization, and sexually transmitted infections control. We found that in 2022, 55% of eligible Ethiopian women were screened for cervical cancer (against the annual target), with roughly half of those with a positive result receiving treatment. Overall, 900,000 (8.4%) of the 10.7 million eligible women in the country underwent screening. The study revealed inadequate implementation of the NCCP strategies toward achieving the WHO's 90-70-90 cervical cancer targets by 2030. A key positive strategy was the involvement of high-ranking government officials in the National Cancer Committee, which aided the NCCP implementation. On the other hand, inadequate political support, funding constraints, suboptimal public messaging, lack of incentives, and inadequate partnership arrangements emerged as important barriers. We recommend that decision-makers intensify coordinated efforts, prioritize dealing with identified challenges and optimizing facilitators, and mobilize additional resources to enhance cervical cancer services in Ethiopia.

**Funding:** The authors received no specific funding for this work.

**Competing interests:** The authors have declared that no competing interests exist.

## Introduction

There were 604, 127 new cervical cancer cases with 341,831 deaths in 2020 [1]. Of these, Africa accounted for 117, 316 incidences. Roughly 76,400 deaths were reported in sub-Saharan Africa in 2018 alone [2]. Ethiopia is similarly afflicted, with cervical cancer being the second most prevalent cancer behind breast cancer and the second leading cause of cancer-related deaths among women [3–5]. In 2020, there were 7, 445 new cases of cervical cancer (out of 54,560 cases in Eastern Africa) and around 5, 338 cervical cancer deaths (out of 36,497 deaths in Eastern Africa) [5]. In Ethiopia, an estimated 3.8% of women in the general population are infected with the cervical human papillomavirus (HPV) 16 and/or 18, with nearly 37 million women aged 15 and above being at risk of getting cervical cancer [5]. Cancer is also a disease that has received little attention from the Ethiopian Government in the past [3] for various reasons, including resource constraints and excessive focus on communicable diseases [6]. Low public awareness, inadequate diagnostic and treatment facilities, a shortage of oncology professionals, and poor referral pathways were the challenges in the fight against cancer [7]. Ethiopia's government established a National Health Policy in 1993, and the country's developmental objectives prioritize health. Through a multisectoral approach in a decentralized three-tier healthcare delivery system (the primary, secondary, and tertiary levels), the strategy sought to increase healthcare accessibility and equity. At the primary level, health posts serve 3,000–5,000 individuals, health centers support 15,000–25,000 persons, and primary hospitals reach 60,000–100,000 people. While 1 to 1.5 million people receive healthcare from secondary-level general hospitals, and tertiary-level specialized hospitals serve 3.5 to 5 million people [6]. The rapid expansion of the private-for-profit sector complements the health system. In 2020, Ethiopia's healthcare spending accounted for 3.48 percent of GDP [8]. GDP per capita was USD825, while health spending per capita averaged USD26, with government health spending accounting for 30.5% and out-of-pocket spending standing at 37.0% in 2021 [8].

According to the World Health Organization's (WHO) 2017 Cancer Resolution, cancer is a serious and expanding public health concern that needs to be given international priority, attention, and funding [9]. To achieve universal health coverage (UHC), nations must establish cancer control frameworks that have links to other national policies and are based on a thorough understanding of the context of the healthcare system. Ethiopia created and executed the National Cancer Control Plan (NCCP) in 2015. The NCCP implementation framework has defined strategies, objectives, interventions, and monitoring indicators to prevent and control cancer. Fig 1 outlines our conceptual framework. A continuum of cancer care requires a coordinated array of actions, beginning with public awareness campaigns, risk identification, and HPV vaccination as fundamental preventive measures. This is followed by early cancer detection and screening, diagnostic and treatment services, and palliative care. In addition, the NCCP mandates that the nation establish an ongoing research and surveillance system [6]. The burden (morbidity and death) of cervical cancer in Ethiopia is still alarmingly high even with the NCCP in effect. This indicates that Ethiopia should accelerate its efforts to meet the WHO's 90-70-90 cervical cancer targets by 2030 [10]. The initiative aims to attain 90% of girls immunized against HPV by the age of 15; 70% of women screened for cervical cancer twice in their lifetime (by the age of 35 and 45); and 90% of women who have invasive cancer or pre-invasive lesions treated. This study examined the barriers and facilitators to implementing Ethiopian NCCP strategies, its connection with other national policies, and target attainment for cervical cancer services in Ethiopia.

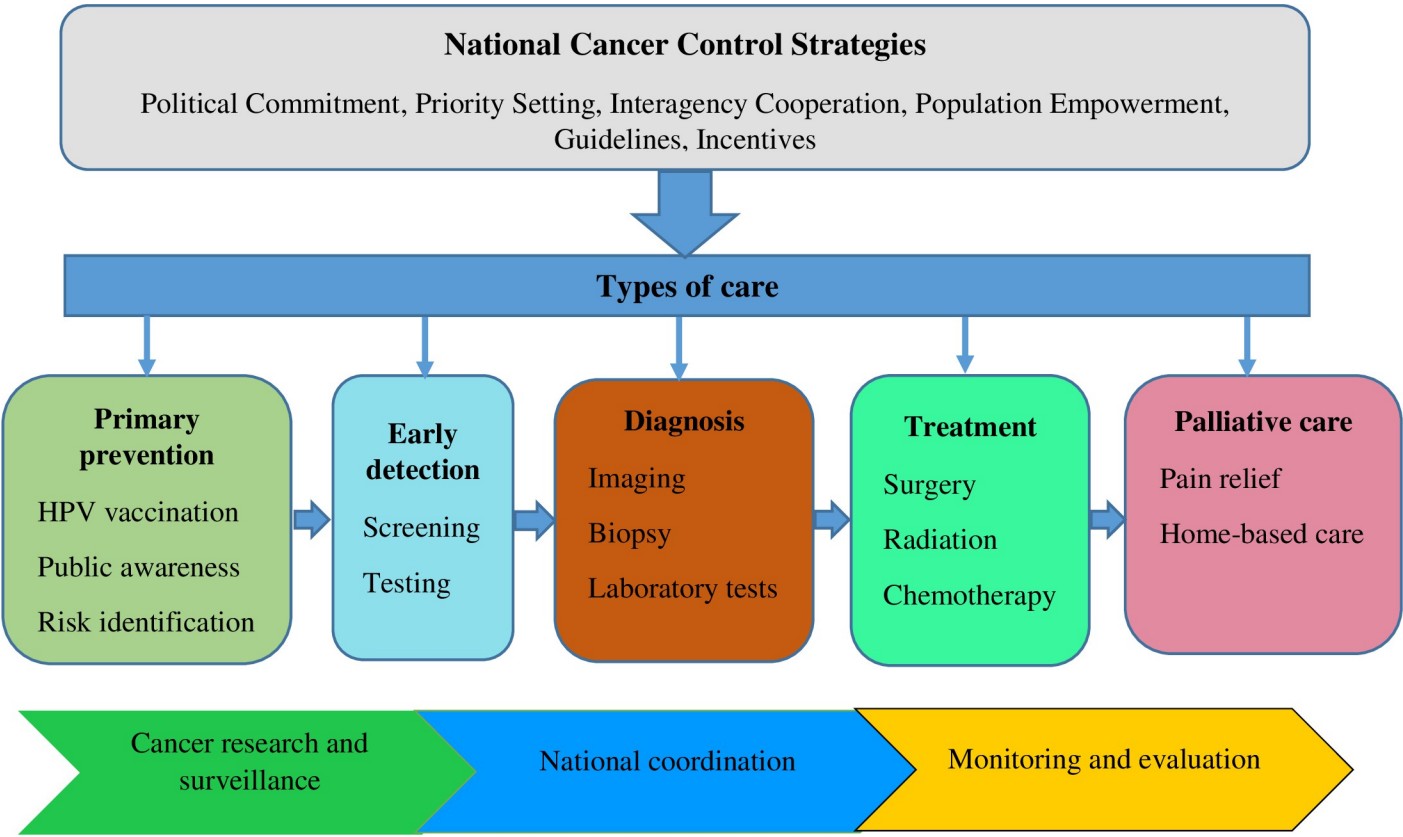

**Fig 1. Conceptual framework: A modification of the Ethiopian national cancer control plan continuum of cancer care.**

## Materials and methods

### Study design and period

The study utilized a qualitative method approach, combining key informant interviews with analyses of information from secondary sources. Key informant interviews were done to explore political commitment, priority setting, interagency cooperation, evidence-based practice, population empowerment, and incentive strategies of the NCCP implementation. The NCCP's implementation status (HPV vaccination, cervical cancer screening, and treatment) and its linkage to key national plans and policies (for cervical cancer crosscutting interventions) were determined by reviewing secondary data. The study was reported in compliance with "the consolidated criteria for reporting qualitative research (COREQ)": a 32-item checklist to assess qualitative studies [11]. The timeframe of data collection was from October 1, 2022, to October 31, 2022.

### Study sites and population

The study sites, study participants, and secondary data sources are depicted in Table 1. The main stakeholders involved in this project were the Ministry of Health, the Addis Ababa City Administration Health Bureau, and development partners.

Table 1. The study sites, study population, and secondary data sources.

| | MOH | AACAHB | Development partners |
|---|---|---|---|
| **Study sites** | • Disease Prevention and Control Directorate<br>• MCH and Nutrition Directorate<br>• Ethiopian Pharmaceuticals Supply Service<br>• Ethiopian Food and Drug Authority | • NCDs Directorate<br>• MCH Directorate | • Centers for Disease Control and Prevention<br>• Family Guidance Association of Ethiopia<br>• Pathfinder International<br>• Wings of Healing<br>• AIDS Healthcare Foundation |
| **Study participants** | • Directors<br>• Cervical cancer experts<br>• EPI officers | • Director<br>• Cervical cancer expert | • Directors<br>• Public health specialists |
| **Secondary data sources** | • EPI and cervical cancer registers<br>• HSTP, NSAP for NCDs, Cervical Cancer Guidelines, NRHS, and Roadmap for Health Extension Program | • EPI and cervical cancer registers | • None |

Key: - AACAHB: Addis Ababa City Administration Health Bureau; EPI: Expanded Program on Immunization; HSTP: Health Sector Transformation Plan; MCH: maternal and child health; MOH: Ministry of Health; NCDs: non-communicable diseases; NRHS: National Reproductive Health Strategy; NSAP: National Strategic Action Plan.

## Sampling

Interviewees were purposively selected from the Ministry of Health, Addis Ababa City Administration Health Bureau, and development partners. To determine cervical cancer program performance, three monitoring indicators (HPV vaccination, screening, and treatment) were selected from the NCCP implementation framework and used [6]. Five key policy documents (Table 1) whose mandates intersect with the NCCP were included in the reviews.

## Data collection procedure

The data collection happened simultaneously using key informant interviews, and a review of secondary data. The principal investigator (corresponding author), a male pharmacist (BPharm, MSc) experienced in qualitative research, conducted face-to-face key informant interviews with fifteen participants in their workplace, using customized guides (S1 File). These included eight directors, two experts on cervical cancer, two EPI officers, and three experts on public health, to explore the barriers and facilitators of the NCCP implementation strategies. The strategies covered in the interviews were population empowerment, evidence-based practice, prioritization, political commitment, partnerships, and incentive programs, modified from the WHO Better Non-communicable Disease Outcomes- Challenges and Opportunities for Health Systems - Assessment Guide [12]. The principal investigator followed "the consolidated criteria for reporting qualitative research (COREQ)": a 32-item checklist (S2 File) [11]. Before the start of the interviews, no relationship was built with the participants. However, participants received formal communication two to three days before the interviews through their respective organizations. During the consent form explanation, the participants were informed of the researcher's motivations for conducting the study. The principal investigator was unbiased, nonjudgmental, and showed due respect to participants. Hence, none of the participants withdrew or declined to take part in the study. For the interviews, only the participants and the lead investigator were present. The interviews were audio-recorded using a smartphone in Amharic (the country's official language) and/or field notes were taken. The data were collected until saturation and labeled with codes. An average of one hour was spent on each interview. The principal investigator reviewed the interviews right after the participant discussions to ascertain if the guides were relevant for addressing the

study questions and whether the responses were sufficient to conclude. Thus, no repeat interviews were carried out.

The corresponding author also conducted secondary data reviews at the Ministry of Health to determine the percent achievement (performance per the annual plan) for HPV vaccination, screening, and treatment in 2022. A checklist was used to extract all data from the Ministry of Health's cervical cancer register reporting the number of women who were screened for the disease and who underwent treatment in the country (for the period covering July 1, 2021, to June 30, 2022). Whereas the national EPI register reports were used to determine the HPV vaccination target attainment in 2022. On the other hand, we reviewed cross-cutting strategies within the NCCP, cervical cancer guidelines, health extension roadmap, health sector transformation plan, strategic plan for NCDs, and reproductive health strategy.

## Data processing and analysis

A full transcription in Amharic, a translation into English, and coding were made by the corresponding author from the audio recordings of the key informant interviews. Since the transcripts were already reviewed for accuracy and completeness, they were not sent back to participants for corrections and/or comments. The English translations of the key informant interview textual data and notes taken during interviews were arranged and categorized in Microsoft Excel, coded, and analyzed into six pre-established themes (political commitment, priority setting, interagency cooperation, evidence-based practice, population empowerment, and incentives), and sixteen sub-thematic areas using a thematic analysis approach. As part of the coding process for qualitative data, labels were systematically given to data segments to represent various contents, ideas, and concepts. The codes were subsequently put together for each of the themes. In addition to establishing consistency between the data presented and the findings, participant quotations were used to illustrate the key findings in the respective themes. Whereas the data on cervical cancer service achievement (including screening, treatment, and HPV vaccination) obtained from national records were checked for completeness and accuracy. The percentages were summarized and presented in the form of figures. On the other hand, the document reviews were cleaned and analyzed using Microsoft Word. Lastly, an interpretation was made for the processed and analyzed qualitative data.

## Ethical consideration

Ethical approvals of the study were obtained from Strathmore University Institutional Scientific and Ethical Review Committee (SU-IERC1373/22), and Addis Ababa City Administration Health Bureau Ethical Clearance Committee (A/A/0024/227). The participants were asked to read and sign a written informed consent form before the interview. Further, the participants received a comprehensive explanation of every question asked during interviews, a part of the study's guidelines. Participants were free to change their minds and discontinue participation at any point during the study. Furthermore, the confidentiality of the collected information was maintained, and no identifiable names were displayed in the study.

## Results

### 1. Key informant interviews on the NCCP strategies

Table 2 shows the six thematic topics, sixteen sub-thematic areas, barriers, and facilitators identified through the key informant interviews on the implementation of NCCP strategies. The presence of the national cancer committee and a strong government commitment to the NCCP implementation were identified during the interviews as important facilitators. On the

**Table 2. Thematic analysis of the barriers and facilitators of the NCCP strategies.**

| Themes | Sub-themes | Barriers | Facilitators |
|---|---|---|---|
| **Political commitment** | Commitment of the government | • Inadequate comprehensive political support | • Dedicated government political leadership<br>• Committed national cancer committee |
| | Stakeholders' attention | • Inadequate attention was given to domestic resource mobilization<br>• Lack of well-organized management of partners | • Government agencies support<br>• Development partners support |
| | Budgeting for cancer | • Shortage of funding | • Development partners support |
| **Priority-setting** | Setting priorities | • Few centers of excellence hospitals<br>• Low in-service training of healthcare providers<br>• Shortage of specialty care (complex diagnostics, chemotherapy, oncology surgery, pathology, and radiotherapy)<br>• Inadequate Electronic Health Records (EHRs), lack of cancer registries, and computer e-referrals.<br>• Low screening, and treatment achievements<br>• Lack of a national population cancer registry and cancer surveillance system | • A decentralized health system<br>• National cancer technical working group<br>• Adequate number of middle-level health workers<br>• Experience of the population cancer registry in Addis Ababa<br>• Development partners support<br>• Regional institutions support |
| | Mobilizing resources | • Poor public and private partnership | • Presence of a strong private health sector |
| | Limitations in funding | • Lack of clearly marked budget | • Development partners support<br>• Presence of community-based health insurance |
| **Interagency cooperation** | Participation in technical working group | • Narrow representation of stakeholders | • Oncology hospitals, universities, research institutes |
| | Multisectoral assistance | • No functional partnership between the public and private sector<br>• Limited partnerships among government agencies<br>• Lack of international partnerships | • Various government ministries support<br>• Availability of different professional associations |
| **Integrating evidence into practice** | Capacity of experts | • A low mix of adequately trained health workforce | • Continuing professional development training by public and private medical schools<br>• MOH Digital Health Activity |
| | Developing and disseminating guidelines | • Lack of cancer standard treatment guidelines | • Specialized hospitals support |
| | Training and monitoring of providers | • Lack of integration of guidelines in formal education<br>• Lack of e-learning modules | • Medical schools support |
| **Population empowerment** | Programs for empowering communities | • Inadequate resources<br>• Insufficient public awareness messages<br>• Low involvement of cervical cancer survivors, religious leaders, traditional healers, traditional leaders, civil societies, and private sector | • Development partners support<br>• Strong private sector<br>• Availability of local TV programs and FM radios<br>• Availability of adequate community health workers<br>• The presence of widespread telecom infrastructure |
| | Patient support efforts | • Lack of peer-to-peer, eHealth, and mHealth supports | • Cancer societies' support<br>• MOH Digital Health Activity |
| **Incentive systems** | Performance-based payment | • Lack of financial incentives | • Recognition certificates, and in-service training |
| | Patients' incentives | • Lack of transportation fees | • Development partners support |
| | Decision-makers challenges | • Lack of resources, sustainability issues, and impact on other programs | • Working through widespread stakeholders |

other hand, budgetary restrictions, limited partnerships, a demand for more political support and its translation to sub-national levels, a shortage of public messaging, and a lack of incentives were the main barriers.

The NCCP's execution was facilitated by key political figures who co-chaired the Ethiopian National Cancer Committee, which coordinates cancer prevention and control efforts. In addition, the presence of the National Cancer Technical Working Group (TWG), the use of

community-based health insurance, the deployment of middle-level oncology professionals, a decentralized health system, the availability of cancer guidelines, and the implementation of the combat cervical cancer (3Cs) initiative were all identified as facilitators.

*"Though the combat cervical cancer (3Cs) initiative was implemented in Ethiopia targeting free screening service for five women in a day in a facility at 1218 health facilities in 800 wore-das (districts) an overall screening achievement was low, 900, 000 women (about 8.4%) out of 10.7 million eligible women (targets) in Ethiopia."* (Key informant interviewee (KII) 1 in MOH)

The interviews uncovered the lack of a regional budget specifically designated for NCDs as well as the inadequate approach to coordinating budgets and planning to implement Health in All Policies (HiAP), which includes cervical cancer services in both the public and private sectors.

*"There was no constant budget set for non-communicable diseases and cervical cancer services in Addis Ababa City Administration Health Bureau. The majority of the budget was allocated by the MOH. The partners and the regional health bureau allocated resources to fill the gaps in the provision of cervical cancer prevention and control services in terms of financing, train-ing, population empowerment, and provision of equipment and supplies."* (KII 9 in AACAHB)

The informants noted that the MOH was working with different government agencies and development partners, who helped in human resources, capacity building, mentorship, sup-portive supervision, public awareness, financing, and provision of health products and tech-nologies. Challenges included low public-private collaboration and poor sustainability of development partners' support. Additionally, it was noted that there were limited collabora-tions between government organizations, which precluded joint planning, execution, and oversight of the cancer services.

*"There was some liaison with Addis Ababa City Administration Education Bureau (AACAEB). However, this area was not adequately exploited and it needs to be strengthened with AACAEB, Women, Youth, and Child Affairs, and Sports Bureau, and others."* (KII 9 in AACAHB)

The informants stated that eHealth, mHealth, and the absence of support groups for cancer patients were obstacles to reaching out to larger population groups. In addition, some partici-pants reported a shortage of in-service training based on cervical cancer guidelines for cancer professionals in private hospitals and rural health extension workers (HEWs).

*"The public awareness for NCDs including cervical cancer in the community was found to be inadequate as HEWs were required to receive integrated refresher training. The training for HEWs was not given in full as it was projected to cost roughly USD 7 million. Therefore, the training was focused on urban settings but not on rural ones."* (KII 1 in MOH)

Informants identified medicines supply chain management issues, primarily due to long procurement lead times, foreign currency shortages, delayed regulatory approvals, and budget-ary limitations. There were up to six formulations (Cisplatin injection, and Paclitaxel injection) registered and five first-line buyers approved by the Ethiopian Food and Drug Authority

(EFDA). Cisplatin injection of 50 mg/50 ml was procured by the Ethiopian Pharmaceuticals Supply Service (EPSS) with a median price of USD1.90 and a median price ratio (MPR) of USD1:7 times international reference prices (IRPs) while Paclitaxel powder injection 6 mg/ml had a median price of USD6.55 and an MPR of USD1:2 times IRPs (the IDA Foundation Electronic Price Indicator, Quarter 2 of 2022).

> *"Hospitals quantify their needs based on the consumption and morbidity methods. They also consider cancer medicines prescribed and procured outside the facility by patients. The facilities staff attended the analysis and validation workshop which was conducted for two days by the MOH, Ethiopian Pharmaceuticals Supply Service, Ethiopian Food and Drug Authority, and other stakeholders. Fifty percent (50%) of the budget for the procurement of cancer medicines was fully subsidized by the MOH based on demand and 50% from the respective hospitals. There was no direct budget transfer by the hospitals but expected to pay after delivery of products."* (KII 5 in EPSS)

Informants saw a lack of financial incentives, such as a monthly fee per beneficiary or a monthly payment for expanded cervical cancer services or achieving specific outcomes related to cervical cancer prevention, diagnosis, or treatment by health providers.

> *"One of the major challenges in our intervention was the lack of motivation of the health providers to elicit and offer the cervical cancer service, it was more of a passive service and on demand of the client than specifically requested and offered for all eligible clients. Unfriendly workflow of the health service: in most of the health facilities, the units providing cervical cancer service were far, less suitable, and less friendly."* (KII 11 in Partners)

### 2. Cervical cancer services achievement at the national level

According to the Ministry of Health's EPI register annual report from 2022, 83% of girls had gotten their second dose of the HPV vaccine by age 15. The percentage achievement (performed versus planned) reported in the cervical cancer register from July 1, 2021, to June 30, 2022, indicated that 55% of women were screened and 53% of women with positive precancer lesions received treatment (Fig 2). In 2022, a total of 900, 000 (8.4%) women were screened out of the eligible 10.7 million in the country.

### 3. The NCCP linkage with other strategies

The connections between the NCCP and other national programs for the prevention and control of cervical cancer are presented in Table 3. Our document review revealed that cross-cutting interventions were put in place for raising public awareness, HPV immunization, sexually transmitted infections control, risk identification, screening, diagnosis, and treatment of cervical cancer in Ethiopia.

## Discussion

The study specifically sought to examine barriers and facilitators to the implementation of the Ethiopian National Cancer Control Plan (NCCP). Despite the Ethiopian Government's political support for the NCCP's implementation, cervical cancer services were far from meeting the WHO's 90-70-90 targets by 2030. This pointed to inadequacies in the execution of NCCP strategies, including inadequate political support, funding constraints, low public messaging,

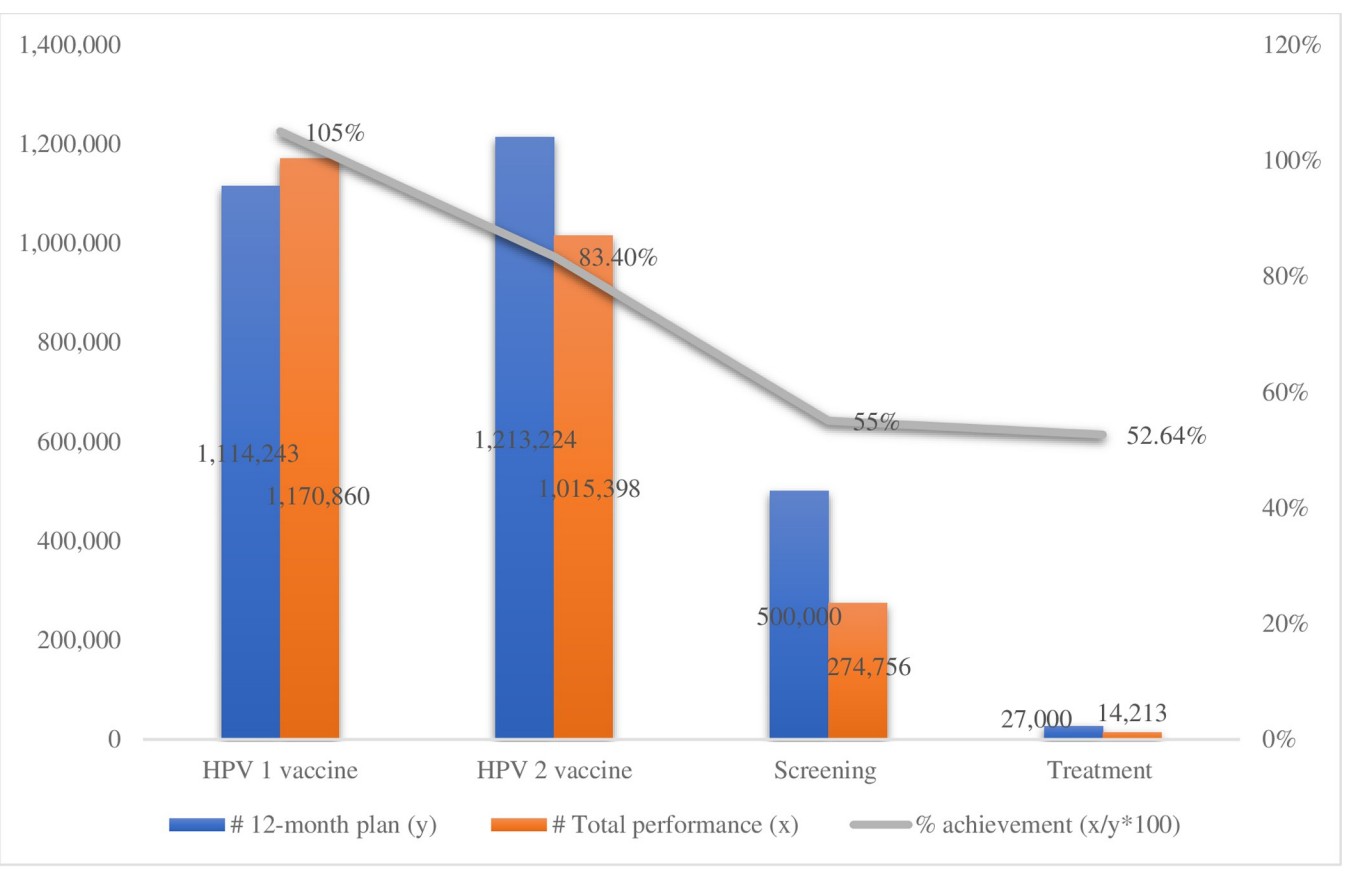

**Fig 2. HPV vaccination, cervical cancer screening, and treatment in Ethiopia, 2022.**

shortage of financial incentives, and inadequate partnerships. All stakeholders are required to enhance coordinated and integrated efforts in terms of resource mobilization, setting priorities, and investing in the prevention and control of cervical cancer in Ethiopia.

According to the key informant interviews, the National Cancer Committee and National Technical Working Group (TWG) in Ethiopia received strong political support for implementing the NCCP through a decentralized health system, which was a major enabler in the battle against cancer. While the political figures led an effort in the fight against cancer, high-profile women provided inadequate support, primarily through public awareness initiatives. To guarantee seamless program execution and efficient resource utilization for cancer prevention and control, the NCCP necessitates the concerted and focused efforts of every key stakeholder [6].

As in previous domains [13, 14], the TWG was crucial in establishing priorities and bringing stakeholders together. Partners provided financial, material, and technical support to help close the resource gap. To scale up and offer equitable services across the nation, the combat cervical cancer (3Cs) initiative, the quarterly TWG meetings, and the MOH yearly review meetings were crucial. Increasing the number of health workers, providing replacement training, and switching up services within a facility were all methods used to lower employee turnover. Nonetheless, healthcare facilities faced challenges with a shortage of oncology specialists, Pap smear tests, HPV DNA tests, Colposcopy, LEEP, CT scans, MRI, and radiotherapy services. Various strategies for scaling up cervical cancer services were reported by previous

**Table 3. The NCCP linkage with other national plans.**

| Guidelines/plans/strategies | Linkage with the NCCP |
|---|---|
| Health Sector Transformation Plan of Ethiopia II (HSTP II) (2020/21-2024/25) | • The main objective of the Ethiopian HSTP II is to increase the health system's responsiveness to attain universal health coverage, which will ultimately lead to an improvement in the population's health status. It aims to use public-private collaboration to raise the number of women aged 30–49 who get screened for cervical cancer from 5% to 40%. |
| National Strategic Action Plan (NSAP) for Control of NCDs in Ethiopia (2014–2016) | • The goal of the NCCP strategies is to further enhance NSAP implementation for better-controlling NCDs. Risk factors and preventative strategies for NCDs are similar. Consequently, the NCCP put forth a planned intervention aimed at reducing exposure to common cancer risk factors, including excessive alcohol intake, tobacco use, physical inactivity, and unhealthy diets. |
| Guideline for cervical cancer prevention and control in Ethiopia (April 2021) | • The NCCP acknowledges and supports the application of guidelines for the prevention and control of cervical cancer. Instructions on increasing public awareness, HPV vaccination, risk identification, screening, early detection, diagnosis, and treatment of cervical cancer are all provided by the NCCP and the guidelines. The initiatives aim to improve the quality of life, lower disability, and raise the odds of survival for cancer patients. Policymakers, health managers, healthcare professionals, educational institutions, and development partners across the nation are intended users of the guidelines. |
| National Reproductive Health Strategy (2016–2020) | • Ethiopia's reproductive health strategy strives to improve the health of women, adolescents, and youngsters by making high-quality services available and accessible in the healthcare system. According to the policy, reproductive organ cancers and sexually transmitted infections (STIs) must be prevented and treated using an integrated approach. Consequently, to address HPV through national immunization programs, the NCCP, STI strategies, and cervical cancer guidelines seek to provide coordinated services. |
| A Roadmap for Optimizing the Ethiopian Health Extension Program (2020–2035) | • Health extension programs (HEPs) are provided by primary healthcare units (PHCUs), which serve a population of 5,000 in the community through illness prevention and control, and family health services including for cervical cancer. Following the objectives of the NCCP, a Roadmap for Optimizing the Ethiopian Health Extension Program was put into place to assist in the achievement of universal health coverage. Health extension workers are responsible for identifying the women who are most at risk for cervical cancer and referring them to public health facilities for screening, in addition to spreading public awareness messaging. |

studies [13, 15, 16]. Another significant issue that requires attention from decision-makers is the absence of in-service training for oncology professionals in the private sector. However, the public sector has made big strides in comparison with the Ethiopian research, which found that in 2014, only 4% of the facilities had staff members who had received in-service training on cancer [17]. Expansion of population-based cancer registry sites has been recommended in the past [4]; however, it continues to be a challenge for decision-makers to make informed and evidence-based policy decisions. Previous research suggested allocating financial and technical resources to the creation of cancer registries to record crucial data, such as the incidence and mortality of cervical cancer and the five-year survival rate of cases of invasive cervical cancer, which contribute to the disease's eventual elimination [13]. All stakeholders are required to address the absence of a cancer surveillance system.

Inadequate funding, lack of foreign exchange, extended regulatory authorization, and a dearth of suppliers for small-scale procurements resulted in extended lead times (8–10

months) for acquiring and distributing cancer medicines by the Ethiopian Pharmaceuticals Supply Service. The Ethiopian Food and Drug Authority enabled a fast-track mechanism for approving cancer medications (within 90 days) that have been prequalified by the WHO to reduce lead times. An Addis Ababa survey found that the mean availability of the lowest-priced generic cancer medicines was 18.8% in the private sector and 56.9% in the public sector [18]. This result differed from the WHO's reported mean availability of generic NCD medications, which was 36% in the public sector and 55% in the private sector [19]. Moreover, the 2012 World Health Assembly mandate of 80% affordable essential medicines available to treat NCDs by 2025 had not yet been met by the strategies for the provision of cancer medicines [20]. As compared to the earlier Ethiopian research [18], our study results demonstrated a better median price ratio for the lowest-priced generic cancer medications. A prior Ethiopian study recommended bolstering supply chain management and increasing resources (through government, private sector, and non-government organizations) to enhance access to medicines for cancer in the healthcare system [21]. Improving access to cancer medications could also benefit from understanding the reform of ART access in the context of HIV/AIDS programs [14].

To create and distribute guidelines for cervical cancer, the Ministry of Health had an adequate number and mix of oncology specialists at the federal level. Compared to results of a 2014 survey [17], the guidelines were disseminated more broadly at healthcare facilities. They were primarily utilized for healthcare workers' in-service training, service delivery, supervision, mentorship, and monitoring and evaluation of activities. Healthcare professionals can benefit from online education such as through MOH Digital Health Activity as an adjunct to traditional classroom instruction to broaden their knowledge, develop their skills, and stay up to date in the field of cancer services.

The main obstacles to increasing the public awareness messaging were determined to be inadequate planning, budgeting, coordination, and engagement among stakeholders. This resulted in messages that were not broad or consistent. Another flaw in the healthcare system was the lack of peer-to-peer, eHealth, and mHealth support for cancer patients and their families. Using digital health solutions such as MOH Digital Health Activity, patients could be empowered to take charge of their care through remote consultations and treatment. On the other hand, there was a paucity of use of traditional healers and religious leaders, who play significant roles in the communities in dispelling myths, providing accurate information, and encouraging cancer-related healthy behavior. Related challenges have been reported in previous studies [7, 16, 17, 22]. All interested parties must be involved in addressing these challenges and implementing a well-thought-out, planned, and well-funded awareness campaign in the nation.

Inadequate performance-based incentives, primarily in the form of finances, prevailed in the health system. This, however, was a recommendation of the Ethiopian Health Sector Transformation Plan II as well as other previous Ethiopian studies [23–25]. Healthcare professionals might not be sufficiently motivated to prioritize cervical cancer services in the absence of financial incentives, particularly if they are dealing with competing demands or scarce resources. The Oncology Care Model also recommends payment incentives [26]. Primary care decision-makers may prioritize looking into offering incentives for health extension workers to identify the most vulnerable women and connect them to healthcare facilities. This may match their interests and increase screening uptake and utilization in the country.

For evidence-based planning, technical support, capacity building, and skills transfer, international cancer control collaborations were necessary. Previous studies reported on a variety of collaborations in the implementation of cancer-control plans [27, 28]. Any opportunities for regional cooperation, like those with the Common Market for Eastern and Southern Africa

(COMESA) and the Intergovernmental Authority on Development (IGAD), could be explored by the national decision-makers. These partnerships could be used to advance collaborative research, knowledge exchange, the establishment of cancer registries and surveillance systems, and capacity building. Moreover, some of the areas that may be considered in the context of regional collaboration in the fight against cancer include the African Comprehensive Cancer Consortium (AC3), which improves cancer treatment and care; African Radiation Oncology Group (AFROG), which addresses the issue of radiation oncology in Africa; African Organization for Research and Training in Cancer (AORTIC), which promotes research projects and training; and African Cancer Registry Network (AFCRN), which improves African cancer registration and surveillance.

More effort is required to improve the national cervical cancer screening attainment rate (annual performance versus plan) of 55% and overall target achievement rate of 8.4% if the WHO goal of 70% is to be met by 2030 [10, 14]. Governments and international organizations were urged by the 2017 World Health Assembly Cancer Resolution to allocate more resources for integrated and coordinated cancer prevention and control, including enhancing screening [9]. Moreover, a "screen and treat" approach is advised by the WHO, in which patients with positive screening results are treated in the same facility [29]. Likewise, the NCCP promotes a "see and treat" approach to cervical screening by educating the public, identifying the most vulnerable women in the community, and linking them with healthcare facilities [6]. However, since the healthcare institutions lacked complete testing and treatment tools, the "screen and treat" method was not entirely realized. Poor referral pathways were identified as a major challenge to cervical cancer prevention in Kenya [22]. To guarantee equitable access, affordability, and sustainability for all, cancer care requires careful planning and efficient utilization of available resources [30].

The document reviews demonstrated the implementation of several cross-cutting cancer initiatives, such as the public-private partnership established by the Ethiopian Health Sector Transformation Plan, which aims to boost the screening rate of eligible women from 5% to 40% (by 2025). Nevertheless, the application of public-private partnerships in the battle against cancer was poor. This corroborates an Ethiopian study that revealed issues such as fragile partnerships between the public and private sectors, mistrust between the two domains, and limited incentives for the private sector [31]. Similarly, poor cooperation between the public and private sectors was one of the factors that affected Tanzania's adoption of cervical cancer screening programs [32]. Other national strategies treat STIs, raise public awareness, support HPV vaccinations, screening, early detection, diagnosis, and treatment programs, address similar risk factors, and use common preventative strategies for controlling and managing NCDs including cancer. These comprehensive approaches are consistent with the NCCP implementation strategies [6].

## Limitations of the study

The lack of resources limited the collection of opinions on the barriers and facilitators of implementation strategies for the Ethiopian National Cancer Control Plan from patients, healthcare providers, the media, professional associations, and cancer societies. Moreover, social desirability bias might have prevented us from gathering more information. Also, if expert panel discussions had been incorporated into the study design, a diverse range of views would have been collected from participants. These might have restricted the comprehensiveness of our study findings.

## Conclusions and recommendations

The Ethiopian National Cancer Control Plan (NCCP) strives to expand equitable cervical cancer services in the country. Our study identified that the NCCP implementation had the necessary political support from prominent national political personalities. Working through the national cancer committee, the national cancer technical working group, the support of development partners, decentralized health services, and implementation of guidelines were additional facilitators in the fight against cervical cancer. In contrast, the NCCP implementation faced several barriers including insufficient multilayered political support, budgetary restrictions, lacking partnerships, poor public messaging, a lack of incentives, and inadequate capacity building of cancer care providers. It was discovered that the national cervical cancer screening rate of 8.4% was far below the WHO's 70% target to be attained by 2030. The NCCP's cross-cutting strategies were adequately incorporated within key national policy documents. Decision-makers in coordination with the private sector, development partners, and non-governmental organizations should allocate more resources and address the identified barriers to NCCP implemenatation strategies taking into account the population's cancer prevention and control needs, and the country's low healthcare capacity. These efforts should aim to eliminate cervical cancer in the next decades, with a set of WHO targets to be reached by 2030.

## Supporting information

**S1 File. Key informants interview guide to explore Ethiopian national cancer control plan strategies implementation.**
(DOCX)

**S2 File. Completed consolidated criteria for reporting qualitative studies (COREQ): A 32-item checklist.**
(DOCX)

**S3 File. Ethiopian national cancer control plan strategies implementation data.**
(ZIP)

**S4 File. Ethiopian national cancer control plan, other policies, and cervical cancer services performance data.**
(ZIP)

## Acknowledgments

We are grateful to health facility managers, the Ministry of Health, the Addis Ababa City Administration Health Bureau, and its sub-cities health offices for the approval of the study and the continuous support provided throughout the study period. We also thank all key informants who provided appropriate information.

## Author Contributions

**Conceptualization:** Kemal Hussein, Francis Wafula, Getnet Mitike Kassie, Gilbert Kokwaro.

**Data curation:** Kemal Hussein.

**Formal analysis:** Kemal Hussein.

**Investigation:** Kemal Hussein.

**Methodology:** Kemal Hussein, Francis Wafula, Getnet Mitike Kassie, Gilbert Kokwaro.

**Supervision:** Francis Wafula, Getnet Mitike Kassie, Gilbert Kokwaro.

**Validation:** Kemal Hussein, Francis Wafula, Getnet Mitike Kassie, Gilbert Kokwaro.

**Writing – original draft:** Kemal Hussein.

**Writing – review & editing:** Francis Wafula, Getnet Mitike Kassie, Gilbert Kokwaro.

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
