## [Decision Letter · Decision Letter 0]

16 Nov 2023

PGPH-D-23-00909

Implementation of the Ethiopian National Cancer Control Plan: An examination of the facilitators and barriers across eight hospitals in Addis Ababa.

Dear Dr. Hussein,

Thank you for submitting your manuscript to PLOS Global Public Health. After careful consideration, we feel that it has merit but does not fully meet PLOS Global Public Health’s publication criteria as it currently stands. Therefore, we invite you to submit a revised version of the manuscript that addresses the points raised during the review process.

This research is of great importance and interest; however, the authors have a lot of work to do to bring the presentation of this manuscript up to standard. There needs to be a clear conceptual framework that is driving the whole research process including analysis of findings. A clear description of the methodology is needed to ensure the research is valid, for example the process of developing interview guides and whether translation, back-translation etc., was done. The authors need to ensure that findings are reported in standard format and should include the COREQ checklist for qualitative work, and the RECORD checklist for the quantitative findings.  The authors can improve the presentation of the quantitative and qualitative results. Preferably, use tables to condense and organize the qualitative findings into themes, subthemes and one supporting quote per subtheme. Further synthesis and triangulation of findings will be needed to show the overarching theme and takeaways messages. As it is now, there is a ot of interesting information but no structure and coherence - it is easy for the reader to get lost in the information. Lastly, get copy-editing assistance to ensure that the overall messaging does not overshadow the importance of the science being presented. 

We look forward to receiving your revised manuscript.

Kind regards,

Beryne Odeny, M.D., MPH, Ph.D

Academic Editor

Journal Requirements:

1. In the ethics statement in the Methods, you have specified that verbal consent was obtained. Please provide additional details regarding how this consent was documented and witnessed, and state whether this was approved by the IRB.

Additional Editor Comments (if provided):

This research is of great importance and interest; however, the authors have a lot of work to do to bring the presentation of this manuscript up to standard. There needs to be a clear conceptual framework that is driving the whole research process including analysis of findings. A clear description of the methodology is needed to ensure the research is valid, for example the process of developing interview guides and whether translation, back-translation etc., was done. The authors need to ensure that findings are reported in standard format and should include the COREQ checklist for qualitative work, and the RECORD checklist for the quantitative findings. The authors can improve the presentation of the quantitative and qualitative results. Preferably, use tables to condense and organize the qualitative findings into themes, subthemes and one supporting quote per subtheme. Further synthesis and triangulation of findings will be needed to show the overarching theme and takeaways messages. As it is now, there is a ot of interesting information but no structure and coherence - it is easy for the reader to get lost in the information. Lastly, get copy-editing assistance to ensure that the overall messaging does not overshadow the importance of the science being presented.

Reviewers' comments:

Reviewer's Responses to Questions

**Comments to the Author**

1. Does this manuscript meet PLOS Global Public Health’s publication criteria? Is the manuscript technically sound, and do the data support the conclusions? The manuscript must describe methodologically and ethically rigorous research with conclusions that are appropriately drawn based on the data presented.

Reviewer #1: No

Reviewer #2: Partly

Reviewer #3: Partly

2. Has the statistical analysis been performed appropriately and rigorously?

Reviewer #1: No

Reviewer #2: No

Reviewer #3: Yes

3. Have the authors made all data underlying the findings in their manuscript fully available (please refer to the Data Availability Statement at the start of the manuscript PDF file)?

Reviewer #1: No

Reviewer #2: Yes

Reviewer #3: Yes

4. Is the manuscript presented in an intelligible fashion and written in standard English?

Reviewer #1: Yes

Reviewer #2: No

Reviewer #3: Yes

5. Review Comments to the Author

Reviewer #1: The subject of cancer prevention and control is always considered an important one because there are lessons to be learned. Unfortunately, this study was not based on public health principles from the onset, hence no true scientific foundation.

The title is too long and requires modification. Suggestion for modification "Predictors of implementation plans for the Ethiopian National Cancer Control programme"

It is believed that the facilitators and barriers are the major predictors however, these predictors must be identified based on scientific principles underpinning the problem phenomenon, Therefore appropriate theoretical and public health principles of prevention and control emerging from the natural history of cancer pathology and trajectory, otherwise any outcome may not be considered valid.

Abstract/Introduction

The thesis of the study appears weak. Emerging questions: What theoretical principles guided the framework developed by the National Cancer Control Plan (NCCP)?

There are no variables as such in the study.

Reviewer #2: This study combines qualitative and quantitative approaches to examine the facilitators and barriers associated with Implementation of the Ethiopian National Cancer Control Plan in eight facilities in Addis Ababa. Using interviews, secondary data review and surveys, for data collection the study attempts to describe the status of implementation and provides recommendations for improvement. The authors apply thematic analysis for the qualitative data to identify and highlight key gaps, and descriptive analysis for the quantitative data to show how facilities performed in the implementation of cancer control.

The authors tackle a critical health issue and their work is much needed in this space. However, the paper is missing major elements of research and manuscript writing. The statement of objective is clear but not consistently addressed throughout. Furthermore, the organization is choppy and the language used is not always clear and unambiguous. This paper could make a contribution to the literature but only if the authors spend a considerable amount of time improving how this work is analyzed and presented.

Reviewer #3: Thank you for the opportunity to review this manuscript. I found the article to be very interesting and informative. I believe the authors are off to a very good start, but recommend some revisions prior to publication, as outlined below:

1. The introduction does a nice job of discussing the country-level context and ends with a strong sentence on the study aim. I found the language in the introduction to be a little unclear, however, and I encourage the authors to review the grammar and sentence structure throughout this section. For example, line 56 "Out-of-pocket expenditure (35%) of health expenditure [8]" is not a complete sentence.

2. In the materials and methods - study design and period section, I recommend that the authors elaborate on the type of mixed-methods research design. I am impressed with the multiple methods used, but I think the authors should clarify the type of design. With the way it is currently written, I would assume it is a convergent parallel design in which data collection happened simultaneously (as opposed to an explanatory sequential or exploratory sequential design). Still, a sentence about this would be helpful.

3. In the materials and methods - data collection procedure and analysis, I recommend that the authors provide more information on their data collection instruments. I see that the instruments are included as supplementary files, but they should be referenced somewhere in the text, just like you would reference a table or figure. Because the instruments are provided, the authors do not need to repeat the questions asked in the instruments, but I was curious about where the questions came from. Citation 11, a WHO document, is cited twice in this section, but I would like to see more detail on how that document was used to inform the interview guide and survey instruments. For example, the interview guide is split into six sections - is this in accordance with how that document is structured? (I could not find the WHO document being cited).

4. I commend the authors for their hard work in consolidating their results across multiple data collection methods. I do think, however, that the results section could be restructured to make the results clearer. Right now, I am not really following a clear storyline, but I can tell the authors have an important story to tell. I would recommend adding subheadings with key themes. For example, in line 112, the authors say that their notes were categorized into six thematic areas for content analysis - what are those six themes? Or in lines 237-239, the authors discuss four key challenges; these challenges could be subheadings. Another way to structure would be to outline each method's respective results and then discuss how the results are integrated in the discussion section. I can tell there are some great findings here, but in its current structure, the key results are not clear to me.

5. I appreciate the authors' use of multiple ways to showcase the data (tables, quotes, etc.), but I encourage the authors to provide more context around the table. For Table 2, what do the numbers mean? Is 1900 the number of planned VIA tests? Is 39.3 a performance score out of 100? And if so, how is performance measured/determined? For Table 3, how was a singular rating for each intervention determined? Was it averaged across the survey respondents, or was this rating given by the authors based on the survey results?

6. The discussion section is very thorough, and I appreciate the additional context to Ethiopia's current policies and systems. If the results section is restructured, I recommend making sure the discussion section follows a similar flow to make the sections cohesive. The title and results suggest diving into barriers and facilitators, but the first paragraph of the discussion section only highlights the key barriers.

I really enjoyed reading this article, and I think it can serve as an example to other countries who wish to examine the facilitators and barriers of implementing their own national cancer control plans. Best of luck in your revision process!

6. PLOS authors have the option to publish the peer review history of their article (what does this mean?). If published, this will include your full peer review and any attached files.

**Do you want your identity to be public for this peer review?** For information about this choice, including consent withdrawal, please see our Privacy Policy.

Reviewer #1: No

Reviewer #2: No

Reviewer #3: No

---

## [Decision Letter · Decision Letter 1]

29 Feb 2024

PGPH-D-23-00909R1

Predictors of Ethiopian National Cancer Control Plan Implementation Strategies: Implications for Cervical Cancer Services in Addis Ababa Hospitals.

Dear Dr. Hussein,

Thank you for submitting your manuscript to PLOS Global Public Health. After careful consideration, we feel that it has merit but does not fully meet PLOS Global Public Health’s publication criteria as it currently stands. Therefore, we invite you to submit a revised version of the manuscript that addresses the points raised during the review process.

While the authors have made substantial revisions, there is need for significant refinement in the execution of the revised version, with a specific focus on presentation of the overall design methodology. Methodological improvements are necessary, and I advocate for a shift from a mixed methods to a qualitative analysis designation, in light of the deficiencies in the quantitative methodology.

We look forward to receiving your revised manuscript.

Kind regards,

Beryne Odeny, M.D., MPH, Ph.D

Academic Editor

Journal Requirements:

Academic editor comments:

-The current revision is an improvement, but it still exhibits some clunkiness, particularly in presenting the quantitative aim and connecting it with the qualitative work. As highlighted in one of the reviews, it might be beneficial to eliminate the quantitative component as it adds more confusion than value to the discourse. A suggestion is to develop a separate paper dedicated to the quantitative aim, providing it with undivided focus and potential standalone significance.

-Title: Please eliminate the term "predictors" from the title and refrain from using it throughout the manuscript due to its connotation of causal inference. Instead, use "barriers and facilitators." For instance, consider a title like "Barriers and facilitators to implementation of Ethiopian NCCP cancer control strategies: implications for cervical cancer services in…"

-Abstract: In the abstract, there is exclusive mention of primary prevention, screening, early diagnosis, treatment, and palliative care, which fall under interventions, not strategies. Strategies, such as political commitment, priority setting, and interagency cooperation, are mentioned in the main document and should be included in the abstract for clarity.

-Throughout the document, maintain the focus of your paper toward barriers to implementation of strategies (implementation strategies). If looking at barriers to interventions, this needs to be clearly stated and delineated in your methods, results, and discussions.

-Methods and results: The study does not follow a mixed methods design and should be characterized as a qualitative analysis. The examination of survey data and secondary sources leans more towards qualitative synthesis than quantitative analysis. Therefore, it is advisable to eliminate references to mixed methods or quantitative analyses, as the quantitative study lacks the components of a traditional study. Notably, there is no specific mention of sampling for quantitative analysis, including sample size calculation, in the methods section. The details regarding the number of individuals sampled for the survey are also unclear. Moreover, the quantitative methodology lacks clarity, with no explicit mention of the variables of interest, outcomes, exposures, or the statistical tests and models employed. In the results section, a table summarizing proportions screened, tested, and treated is presented. However, it's important to note that this is not the outcome of quantitative or statistical analyses; rather, it constitutes a descriptive presentation of proportions without statistical inference. In essence, the study aligns more with qualitative research or a narrative synthesis of data from secondary sources and registers, providing a descriptive summary akin to regular district or national health reports.

-Other comments: Avoid mentioning specific political personalities in the text like the First lady, to avoid the appearance of political agenda/ manifesto; use broader terms like "political figures," "key female decision makers," or "influential people." Maintain objectivity throughout the study, providing objective and citable statements when attributing influence to individuals. If unable to substantiate claims of someone being greatly influential, consider refraining from such phrases to ensure the study's credibility.

Additional Editor Comments (if provided):

The current revision is an improvement, but it still exhibits some clunkiness, particularly in presenting the quantitative aim and connecting it with the qualitative work. As highlighted in one of the reviews, it might be beneficial to eliminate the quantitative component as it adds more confusion than value to the discourse. A suggestion is to develop a separate paper dedicated to the quantitative aim, providing it with undivided focus and potential standalone significance.

Title: Please eliminate the term "predictors" from the title and refrain from using it throughout the manuscript due to its connotation of causal inference. Instead, use "barriers and facilitators." For instance, consider a title like "Barriers and facilitators to implementation of Ethiopian NCCP cancer control strategies: implications for cervical cancer services in…"

Abstract: In the abstract, there is exclusive mention of primary prevention, screening, early diagnosis, treatment, and palliative care, which fall under interventions, not strategies. Strategies, such as political commitment, priority setting, and interagency cooperation, are mentioned in the main document and should be included in the abstract for clarity.

Throughout the document, maintain the focus of your paper toward barriers to implementation of strategies (implementation strategies). If looking at barriers to interventions, this needs to be clearly stated and delineated in your methods, results, and discussions.

Methods and results: The study does not follow a mixed methods design and should be characterized as a qualitative analysis. The examination of survey data and secondary sources leans more towards qualitative synthesis than quantitative analysis. Therefore, it is advisable to eliminate references to mixed methods or quantitative analyses, as the quantitative study lacks the components of a traditional study. Notably, there is no specific mention of sampling for quantitative analysis, including sample size calculation, in the methods section. The details regarding the number of individuals sampled for the survey are also unclear. Moreover, the quantitative methodology lacks clarity, with no explicit mention of the variables of interest, outcomes, exposures, or the statistical tests and models employed. In the results section, a table summarizing proportions screened, tested, and treated is presented. However, it's important to note that this is not the outcome of quantitative or statistical analyses; rather, it constitutes a descriptive presentation of proportions without statistical inference. In essence, the study aligns more with qualitative research or a narrative synthesis of data from secondary sources and registers, providing a descriptive summary akin to regular district or national health reports.

Other comments: Avoid mentioning specific political personalities in the text like the First lady, to avoid the appearance of political agenda/ manifesto; use broader terms like "political figures," "key female decision makers," or "influential people." Maintain objectivity throughout the study, providing objective and citable statements when attributing influence to individuals. If unable to substantiate claims of someone being greatly influential, consider refraining from such phrases to ensure the study's credibility.

Reviewers' comments:

Reviewer's Responses to Questions

**Comments to the Author**

1. If the authors have adequately addressed your comments raised in a previous round of review and you feel that this manuscript is now acceptable for publication, you may indicate that here to bypass the “Comments to the Author” section, enter your conflict of interest statement in the “Confidential to Editor” section, and submit your "Accept" recommendation.

Reviewer #2: All comments have been addressed

Reviewer #3: (No Response)

2. Does this manuscript meet PLOS Global Public Health’s publication criteria? Is the manuscript technically sound, and do the data support the conclusions? The manuscript must describe methodologically and ethically rigorous research with conclusions that are appropriately drawn based on the data presented.

Reviewer #2: Partly

Reviewer #3: Partly

3. Has the statistical analysis been performed appropriately and rigorously?

Reviewer #2: No

Reviewer #3: Yes

4. Have the authors made all data underlying the findings in their manuscript fully available (please refer to the Data Availability Statement at the start of the manuscript PDF file)?

Reviewer #2: Yes

Reviewer #3: (No Response)

5. Is the manuscript presented in an intelligible fashion and written in standard English?

Reviewer #2: Yes

Reviewer #3: Yes

6. Review Comments to the Author

Reviewer #2: The authors have made improvements to the manuscript. However, the storyline is still not clear enough and the writing still requires close editing. As is, the paper is still disjointed. There are multiple data sources and results that are not providing a cohesive main message and certain parts of the results don't seem to inform each other. It is also unclear what the denominators in the results are. I would recommend excluding the quantitative data out. The comparison of policies could use more nuanced reporting and better contextualization in the introduction and methods.

Reviewer #3: The Authors did a wonderful job addressing the reviewers comments in detail. The one aspect that I still feel needs revision is the descrition/use of mixed methods. If the authors would like to stick with the description of convergent parallell design, then the results need to be integrated in some way. Do the qualitative and quantitative results relate to one another? Do they confirm or disprove one another? Did any of the qualitative results provide context to the quantitative findings? OR if the authors would like to keep the results and discussion as is, then I recommend saying that multiple methods (quantiative and qualitative) were used, instead of mixed-methods.

7. PLOS authors have the option to publish the peer review history of their article (what does this mean?). If published, this will include your full peer review and any attached files.

**Do you want your identity to be public for this peer review?** For information about this choice, including consent withdrawal, please see our Privacy Policy.

Reviewer #2: No

Reviewer #3: No

---

## [Editor Report · Decision Letter 2]

18 Apr 2024

PGPH-D-23-00909R2

Barriers and facilitators to implementation of the Ethiopian National Cancer Control Plan strategies: Implications for cervical cancer services in Ethiopia.

Dear Dr. Hussein,

Thank you for submitting your manuscript to PLOS Global Public Health. After careful consideration, we feel that it has merit but does not fully meet PLOS Global Public Health’s publication criteria as it currently stands. Therefore, we invite you to submit a revised version of the manuscript that addresses the points raised during the review process.

We commend you for making the requested changes in the manuscript. A few other issues need to be revised before we proceed.

We look forward to receiving your revised manuscript.

Kind regards,

Beryne Odeny, M.D., MPH, Ph.D

Academic Editor

Journal Requirements:

Additional Editor Comments (if provided):

Thank you for your extensive efforts in revising and restructuring this manuscript.

A few minor points to address before we proceed are as follows:

- In the abstract and throughout, instead of referring to “influential females/ women,” please broadly refer to “lack of political support.”

- Please include the COREQ guideline checklist for qualitative studies in your files and ensure all the requirements of the checklist are met.

- In the Methods, please indicate that the reporting of the study was done in compliance with “the Consolidated criteria for reporting qualitative research (COREQ)” guidelines for reporting qualitative studies.

- Line 257, please remove “other” in “other high-profile women”

- Under limitations, please include limitations that are inherent to the study design itself.

- The conclusion, as it is, reads like a truncation of the results and discussion section. Please synthesize and condense into a concise summary (reduce by half) that ties the themes and takeaways into one or two recommendations for a thought-provoking conclusion.

---

## [Editor Report · Decision Letter 3]

20 May 2024

PGPH-D-23-00909R3

Barriers and facilitators to implementation of the Ethiopian National Cancer Control Plan strategies: Implications for cervical cancer services in Ethiopia.

Dear Dr. Hussein,

Thank you for submitting your manuscript to PLOS Global Public Health. After careful consideration, we feel that it has merit but does not fully meet PLOS Global Public Health’s publication criteria as it currently stands. Therefore, we invite you to submit a revised version of the manuscript that addresses the points raised during the review process.

Please submit your COREQ checklist as part of the supplementary files, not as an email attachment. In the checklist, please refer to headings/ sub-headings and paragraph numbers - do not refer to line numbers as they may change during production of the paper.

We look forward to receiving your revised manuscript.

Kind regards,

Beryne Odeny, M.D., MPH, Ph.D

Academic Editor

Journal Requirements:

1. In the ethics statement in the Methods, you have specified that verbal consent was obtained. Please provide additional details regarding how this consent was documented and witnessed, and state whether this was approved by the IRB.

Additional Editor Comments (if provided):

-Please upload your completed COREQ checklist as a supplementary file. Please refer to headings/ sub-headings and paragraph numbers  - do not refer to line numbers as they may change during production of the paper.
---

## [Editor Report · Decision Letter 4]

7 Jun 2024

PGPH-D-23-00909R4

Barriers and facilitators to implementation of the Ethiopian National Cancer Control Plan strategies: Implications for cervical cancer services in Ethiopia.

Dear Dr. Hussein,

Thank you for submitting your manuscript to PLOS Global Public Health. After careful consideration, we feel that it has merit but does not fully meet PLOS Global Public Health’s publication criteria as it currently stands. Therefore, we invite you to submit a revised version of the manuscript that addresses the points raised during the review process.

We look forward to receiving your revised manuscript.

Kind regards,

Miquel Vall-llosera Camps

Staff Editor

Journal Requirements:

1. In the ethics statement in the Methods, you have specified that verbal consent was obtained. Please provide additional details regarding how this consent was documented and witnessed, and state whether this was approved by the IRB.

Additional Editor Comments:

Thank you for your attention to our previous requests. We have on additional request, please upload your ethics approval documents (original document and translated version) to our file inventory.
---

## [Editor Report · Decision Letter 5]

28 Jun 2024

Barriers and facilitators to implementation of the Ethiopian National Cancer Control Plan strategies: Implications for cervical cancer services in Ethiopia.

PGPH-D-23-00909R5

Dear Mr. Hussein,

We are pleased to inform you that your manuscript 'Barriers and facilitators to implementation of the Ethiopian National Cancer Control Plan strategies: Implications for cervical cancer services in Ethiopia.' has been provisionally accepted for publication in PLOS Global Public Health.

Best regards,

Julia Robinson

Executive Editor